# Domain 2 of Hepatitis C Virus Protein NS5A Activates Glucokinase and Induces Lipogenesis in Hepatocytes

**DOI:** 10.3390/ijms23020919

**Published:** 2022-01-14

**Authors:** Laure Perrin-Cocon, Cindy Kundlacz, Clémence Jacquemin, Xavier Hanoulle, Anne Aublin-Gex, Marianne Figl, Jeremy Manteca, Patrice André, Pierre-Olivier Vidalain, Vincent Lotteau, Olivier Diaz

**Affiliations:** 1CIRI, Centre International de Recherche en Infectiologie, VIRal Infection Metabolism Immunity Team, University of Lyon, Inserm, U1111, Université Claude Bernard Lyon 1, CNRS, UMR5308, ENS de Lyon, 21 Avenue Tony Garnier, F-69007 Lyon, France; laure.perrin@inserm.fr (L.P.-C.); cindy.kundlacz@ens-lyon.fr (C.K.); clemence.jacquemin@inserm.fr (C.J.); anne.aublin-gex@inserm.fr (A.A.-G.); marianne.amandine.figl@gmail.com (M.F.); jeremy.manteca@gmail.com (J.M.); patrice.andre@inserm.fr (P.A.); pierre-olivier.vidalain@inserm.fr (P.-O.V.); vincent.lotteau@inserm.fr (V.L.); 2CNRS ERL9002-BSI-Integrative Structural Biology, 1 rue du Professeur Calmette, F-59019 Lille, France; xavier.hanoulle@univ-lille1.fr; 3Université de Lille, INSERM, CHU Lille, Institut Pasteur de Lille, U1167-RID-AGE-Risk Factors and Molecular Determinants of Aging-Related Diseases, 1 rue du Professeur Calmette, F-59019 Lille, France

**Keywords:** hepatitis C virus, NS5A, glycolysis, glucokinase, lipogenesis, human lipoprotein, central carbon metabolism, hepatocyte

## Abstract

Hepatitis C virus (HCV) relies on cellular lipid metabolism for its replication, and actively modulates lipogenesis and lipid trafficking in infected hepatocytes. This translates into an intracellular accumulation of triglycerides leading to liver steatosis, cirrhosis and hepatocellular carcinoma, which are hallmarks of HCV pathogenesis. While the interaction of HCV with hepatocyte metabolic pathways is patent, how viral proteins are able to redirect central carbon metabolism towards lipogenesis is unclear. Here, we report that the HCV protein NS5A activates the glucokinase (GCK) isoenzyme of hexokinases through its D2 domain (NS5A-D2). GCK is the first rate-limiting enzyme of glycolysis in normal hepatocytes whose expression is replaced by the hexokinase 2 (HK2) isoenzyme in hepatocellular carcinoma cell lines. We took advantage of a unique cellular model specifically engineered to re-express GCK instead of HK2 in the Huh7 cell line to evaluate the consequences of NS5A-D2 expression on central carbon and lipid metabolism. NS5A-D2 increased glucose consumption but decreased glycogen storage. This was accompanied by an altered mitochondrial respiration, an accumulation of intracellular triglycerides and an increased production of very-low density lipoproteins. Altogether, our results show that NS5A-D2 can reprogram central carbon metabolism towards a more energetic and glycolytic phenotype compatible with HCV needs for replication.

## 1. Introduction

Viruses hijack host cell metabolism to promote energy production and synthesis of biomolecules that are necessary for their replication. Interactions between viruses and metabolic fluxes are increasingly studied, which holds promise for the development of original host-targeted antiviral drugs [1,2,3,4,5,6]. Viruses’ capacity to reorient cellular metabolism is best illustrated by hepatitis C virus (HCV) infection that stimulate the synthesis of neutral lipids in hepatocytes to promote the formation of replication complexes at the interface with cytosolic lipid droplets. This also allows HCV to use the lipoprotein synthesis pathway for the production of low-density triglyceride (TG)-rich virions called lipo-viral-particles (LVPs) that are essential to viral spreading [7,8,9,10]. Whereas acute HCV infection is generally asymptomatic and is followed by spontaneous viral clearance in approximately 25% of individuals, in chronically infected patients, this metabolic rewiring contributes to pathological evolution towards steatosis, cirrhosis and hepatocellular carcinoma (HCC) [11]. Studies suggest that several molecular mechanisms lead to an increased lipogenesis and lipid accumulation in infected hepatocytes. HCV core protein was reported to interact with host lipid droplets, thus dysregulating lipid turnover and promoting LVP assembly [12,13]. Additionally, HCV infection induces the activation of Sterol Regulatory Element-Binding Protein (SREBP), the transcription factor responsible for lipogenesis [14,15]. Downregulation of carnitine palmitoyltransferase expression may additionally increase steatosis by inhibition of beta-oxidation [16]. The modulation of Microsomal Triglyceride Transfer Protein (MTTP) by HCV is also involved in the accumulation of intracellular lipid droplets and controls LVP assembly [17,18,19]. Finally, the consumption rate of simple metabolites such as glucose increases in order to feed lipogenic pathways and meet viral needs [20].

These observations highlight the necessity for HCV to interfere with lipid metabolism but the lack of a metabolically competent cell line mimicking hepatic lipogenesis and lipoprotein synthesis has significantly hampered our understanding of this phenomenon [8,15,19,21]. Indeed, the in vitro assembly of bona fide infectious LVPs was only observed in primary human hepatocytes that are functional for the production of very low-density lipoproteins (VLDLs). The HCC cell line Huh7 that is widely used to study HCV replication in vitro is defective for this function, similar to other HCC cell lines such as HepG2 [21]. Furthermore, Huh7 cells exhibit a highly glycolytic profile and have lost the capacity of glycogen storage. Consequently, specific interactions between HCV components, glycolysis and lipid metabolism may have been overlooked because metabolically defective cellular models were used.

Hexokinases control the first rate-limiting step of glucose catabolism by phosphorylating glucose to glucose-6-phosphate (G6P), fueling glycolysis as well as glycogen, pentose phosphate and triglyceride synthesis. The human genome contains five genes encoding distinct hexokinase isoenzymes, named HK1, HK2, HK3, HK4 (also named GCK for glucokinase) and HKDC1, with distinct enzymatic kinetics and tissue distributions. While HK2 is active at low glucose concentration, GCK has an allosteric catalytic comportment with a weak activity at low glucose concentration and a strong activity at higher concentration. Thus, GCK contributes to the regulation of glycemia by controlling hepatic glycolytic activity. During carcinogenesis of hepatocytes, GCK is replaced by HK2 so that HCC cell lines express HK2 instead of GCK. To circumvent this limitation, we recently developed a metabolically active Huh7-derived cell line (Huh7-*GCK^+^/HK2^−^*) by replacing the cancer-type hexokinase HK2 by GCK [22]. We restored the expression of GCK and knocked-down HK2 in Huh7 to generate the Huh7-*GCK^+^/HK2^−^* cell line. We observed that HK2 knockdown and GCK re-expression rewired central carbon metabolism (CCM), stimulating mitochondrial respiration and restoring essential metabolic functions of normal hepatocytes such as lipogenesis, VLDL secretion and glycogen storage [22]. Altogether, this makes the novel Huh7-*GCK^+^/HK2^−^* cell line a functionally relevant model to study HCV interaction with glycolytic, glycogenic and lipogenic pathways.

To rewire cellular metabolism and favor viral replication, viruses can modulate the expression or the activity of enzymes controlling specific metabolic pathways [23,24,25]. However, most of the molecular mechanisms responsible for these metabolic shifts are still poorly understood. The large-scale mapping of virus–host interactions has unraveled many potential interactions between viral and cellular proteins, including metabolic enzymes. By analyzing these interactomes, key metabolic pathways for viral replication were identified [26,27,28,29], but only a few studies have specifically described interactions between viral proteins and host enzymes that directly modulate their activity [30,31,32,33]. One example is the direct interaction between the NS5A protein of HCV (HCV-NS5A) and HK2. This viral protein is composed of three domains (D1, D2 and D3, see Appendix A). Whereas D1 is highly structured and interacts with viral RNA, D2 and D3 are natively unfolded and involved in multiple functions by interacting with several host proteins [34]. We have shown that the interaction between HCV-NS5A and HK2 results in the modulation of HK2 catalytic parameters enhancing the glycolytic activity of Huh7 cells [33]. A limitation to this work was that normal hepatocytes express GCK and not HK2, and that glycogenesis, lipogenesis and VLDL secretion are profoundly altered in Huh7. In the present study, we took advantage of the novel Huh7-*GCK^+^/HK2^−^* cell line to study the consequences of HCV-NS5A expression on these essential metabolic functions and its direct impact on GCK activity. 

## 2. Results

### 2.1. Domain 2 of HCV-NS5A Increases Glycolysis in Huh7-GCK^+^/HK2^−^ Cells

We first analyzed the consequences of HCV-NS5A expression on glycolytic parameters of Huh7-*GCK^+^/HK2^−^* cells. Figure 1A,B show that HCV-NS5A increases glucose consumption and lactate secretion, reflecting a stimulated glycolysis in the presence of HCV-NS5A. We previously reported that the domain 2 of HCV-NS5A (NS5A-D2) was sufficient to enhance HK2 activity in Huh7 cells [33]. We thus asked whether this viral domain alone was able to boost glycolysis in GCK-expressing cells. Huh7-*GCK^+^/HK2^−^* cells were transduced with a lentiviral vector to stably express NS5A-D2 (Appendix A). Glucose consumption and lactate secretion by Huh7-*GCK^+^/HK2^−^* and Huh7-*GCK^+^/HK2^−^*-NS5A-D2 cells was then compared over time (Figure 1C). Similar to cells expressing full-length HCV-NS5A, cells transduced for NS5A-D2 consumed more glucose and secreted more lactate. Thereby, NS5A-D2 is sufficient to stimulate glycolysis in cells expressing GCK.

To further characterize the effect of NS5A-D2 expression on the glycolytic rate of Huh7-*GCK^+^/HK2^−^* cells, we quantified the extracellular acidification rate (ECAR) with an Agilent Seahorse XF Analyzer [35]. Because glycolysis and mitochondria-derived CO_2_ are the two main contributors to ECAR, we precisely determined the proton efflux rate specific to glycolysis (glycoPER), using the Glycolytic Rate Assay [36]. We first measured the basal glycolytic activity, and then inhibited complex I and III of the mitochondrial respiratory chain to block respiration and induce a compensatory glycolysis maintaining energy production (Figure 1D,E). We confirmed that basal glycolysis is higher in Huh7-*GCK^+^/HK2^−^* cells expressing NS5A-D2 compared to control cells (Figure 1D). Blocking the mitochondrial respiratory chain increased glycolysis in both cell lines to maintain intracellular ATP production. However, the compensatory glycolysis was higher in NS5A-D2 expressing cells compared to control cells, indicating an enhanced glycolytic capacity of the cells in the presence of NS5A-D2 (Figure 1E). Altogether, these observations indicate that both basal and compensatory glycolysis are enhanced when NS5A-D2 is expressed in Huh7-*GCK^+^/HK2^−^* cells. 

### 2.2. NS5A-D2 Controls Lipid and Glycogen Metabolism in Huh7-GCK^+^/HK2^−^ Cells

One major function of the liver is the redistribution of lipids in the body through TG-rich lipoproteins that are secreted by hepatocytes. Lipoproteins are assembled from neutral lipids stored into intracytoplasmic lipid droplets. These neutral lipids originate either from the recycling of lipids from the bloodstream or from de novo lipogenesis resulting from glucose catabolism. We previously reported that Huh7-*GCK^+^/HK2^−^* cells have restored lipogenesis and lipoprotein secretion capacities [22]. Since NS5A-D2 expression stimulated glycolysis in these cells (Figure 1), we looked for downstream consequences of this activated glycolysis on lipid metabolism. Interestingly, cells expressing NS5A-D2 secreted more apolipoprotein B (ApoB) (Figure 2A), indicative of an increased production of ApoB+ lipoproteins. Analysis of the ApoB distribution into density gradients of iodixanol confirmed that secreted ApoB was in density fractions ranging from 0.99 to 1.09 g/mL, which is characteristic of very-low density lipoproteins (VLDL) (Figure 2B). The density of the secreted lipoproteins was unchanged by NS5A-D2 expression, indicating that despite an overall increase in lipoprotein secretion, there is no major change in their lipid content. In parallel, intracellular levels of TG were higher in cells expressing NS5A-D2 as determined by biochemical assay and intracellular labeling (Figure 2C,D). Together, these data indicate that stimulation of glycolysis by NS5A-D2 translates into an activated lipid metabolism favoring TG and VLDL production in Huh7-*GCK^+^/HK2^−^* cells.

Hepatocytes also regulate glycemia through uptake and storage of glucose in the form of glycogen. Glycogen then serves as a source of glucose for neoglucogenesis during interprandial periods. We thus investigated whether the expression of NS5A-D2 had an effect on glycogen synthesis. As shown in Figure 2E, Huh7-*GCK^+^/HK2^−^* cells expressing NS5A-D2 showed lower levels of glycogen compared to control cells. In addition, glycogen phosphorylase activity was increased, which is consistent with a higher rate of glycogen degradation in cells expressing NS5A-D2 (Figure 2F,G). Collectively, these observations indicate that NS5A-D2 stimulates glycolysis by increasing glucose consumption and glycogen degradation to support TG accumulation and VLDL secretion.

### 2.3. Effect of NS5A-D2 on Mitochondrial Respiration

Mitochondria are key metabolic and signaling hubs that play an essential role in the metabolic flexibility of hepatocyte and impaired mitochondrial functions contribute to chronic liver diseases associated with obesity, type 2 diabetes and steatosis. Because some of these features are observed during the natural history of chronic HCV infection, we asked if the modulation of hepatocyte metabolism by NS5A-D2 was associated with specific mitochondrial dysfunctions. We measured the oxygen consumption rate (OCR) of cells expressing or not NS5A-D2 with an Agilent Seahorse XF Analyzer (Figure 3A) and showed that their basal respiration was not significantly different (Figure 3B). By addition of inhibitors that are specific to successive steps of the respiratory chain (Mitochondrial Stress Test; Agilent), we also determined the fraction of oxygen that is used for ATP production, the maximal respiration rate, and the spare respiration capacity (Figure 3C). The oxygen consumption profile of Huh7-*GCK^+^/HK2^−^* expressing NS5A-D2 significantly differed from control Huh7-*GCK^+^/HK2^−^* cells by a reduced spare respiration capacity (Figure 3A,C). These results revealed that NS5A-D2 not only stimulates glycolysis but also decreases the energetic plasticity of these cells at the mitochondrion level.

To further explore the impact of NS5A-D2 on energetic metabolism, OCR and ECAR measurements were combined to draw the energy map of the cells (Figure 3D and Appendix A). In standard culture conditions, cells are grown in hyperglycemic conditions (25 mM glucose). Under these conditions, Huh7-*GCK^+^/HK2^−^* cells expressing NS5A-D2 were more energetic than control cells, with especially a higher glycolytic rate. GCK behaves as an allosteric enzyme to adapt liver response to glycemia and is poorly active at normal glycemia. We therefore explored the effect of NS5A-D2 on energetic metabolism at lower glucose concentration matching interprandial periods (5 mM glucose). As expected from the allosteric property of GCK, Huh7-*GCK^+^/HK2^−^* cells were less glycolytic at 5 mM than at 25 mM of glucose. Interestingly, the enhancer effect of NS5A-D2 on the energetic phenotype of the cells was also observed at 5 mM, fully compensating the naturally weak activity of GCK at low glucose concentration. Consequently, Huh7-*GCK^+^/HK2^−^* cells expressing NS5A-D2 were even more energetic at 5 mM of glucose than parental cells at 25 mM of glucose. This indicates that the overall metabolic reprogramming induced by NS5A-D2 is independent of glucose concentration, suggesting that NS5A-D2 may directly interfere with enzymatic characteristics of GCK.

### 2.4. NS5A-D2 Interacts with GCK

We previously observed that HCV-NS5A interacts with HK2 through its domain 2 and modulates its activity [33]. To determine if NS5A also interacts with GCK, we co-expressed the two proteins with specific tags in HEK293T cells and tested their interaction by co-immunoprecipitation. Triosephosphate isomerase 1 (TPI1) was used here as a negative control of the interaction. As shown in Figure 4A, GCK but not TPI1 co-purified with NS5A, thus demonstrating an interaction between NS5A and GCK. We then confirmed this interaction by NanoLuc two-hybrid (N2H), a split-luciferase complementation assay [37]. The two complementary fragments F1 and F2 of the bioluminescent enzyme NanoLuc were fused to the NS5A full length (NS5A-fl) or NS5A-D2, and GCK, respectively, and co-expressed in HEK293T cells. The interaction-dependent luciferase activity confirmed the binding of NS5A-fl or NS5A-D2 to GCK (Figure 4B).

### 2.5. NS5A-D2 Modifies GCK Enzymatic Activity

To determine whether HCV-NS5A binding modulates GCK activity, we analyzed catalytic parameters of the purified enzyme in the absence or presence of HCV-NS5A or its domains. Recombinant human GCK, HCV-NS5A full length (NS5A-fl), the 3 domains of NS5A and HCV-Core were produced in *Escherichia coli* and purified, as previously described [38,39]. The HCV-core protein was used as a negative control. Synthesis of glucose-6-P by the enzyme was quantified by the coupling of HK activity to glucose-6-phosphate dehydrogenase activity and spectrophotometric detection of NADP+ reduction occurring during this second enzymatic step. An increase in GCK activity was observed in the presence of NS5A-fl or NS5A-D2, indicating that NS5A and its D2 domain directly stimulate GCK activity (Figure 5A). HCV-Core had no effect in the same conditions, neither NS5A-D1 nor NS5A-D3 (Figure 5A). 

To better understand how NS5A-D2 can interfere with GCK catalytic parameters, we determined the enzyme velocity according to substrate concentration and established Michaelis-Menten saturation curves (Figure 5B). Results showed that maximum velocity (Vmax) increased by 31% in the presence of NS5A-D2. Lineweaver–Burk representation revealed that, in the presence of NS5A-D2, the exponential curve characteristic of enzyme cooperativity was linearized (Figure 5C), indicating a loss of kinetic cooperativity in the presence of NS5A-D2. Allostery in monomeric single-site enzymes such as GCK is a kinetic phenomenon characterized by the Hill coefficient that describes the cooperativity of a ligand binding to the enzyme [40]. We determined the Hill coefficient of GCK in the absence or presence of NS5A-D2. As previously described, the Hill coefficient of purified GCK was 1.63 (Appendix A and [40]). In the presence of NS5A-D2, the Hill coefficient decreased to 1.30, indicating a lower cooperativity of GCK for glucose. This suggests that NS5A-D2 enhances GCK activity at low concentrations of glucose. 

## 3. Discussion

NS5A-D2 was previously described as an interactor and modulator of HK2 [33]. However, this hexokinase isoenzyme is expressed in HCC cells whereas normal hepatocytes express the hexokinase GCK. With this present study, it is now shown that the D2 domain of HCV-NS5A can also reprogram the metabolism of hepatocytes by interacting with the liver specific hexokinase GCK. At the molecular level, this interaction increases GCK activity by altering its kinetic parameters and allosteric regulation and this has important consequences extending beyond glycolysis. Indeed, the presence of NS5A-D2 in GCK-expressing cells perturbs their glycolytic behavior, modifying their respiratory capacity and redirecting the CCM towards triglyceride synthesis and VLDL secretion. Altogether, these results provide new insight into mechanisms responsible for the metabolic reprogramming of HCV-infected hepatocytes. These data demonstrate that NS5A through its D2 domain provides a better cellular environment for HCV replication and spreading. 

HCV infection only occurs naturally in humans and experimentally in chimpanzees where HCV replicates exclusively in hepatocytes. Although this narrow cellular tropism is not fully understood, it can be attributed for one part to specific metabolic functions of hepatocytes, and primarily their capacity to synthetize TG and secrete VLDL. Chronic HCV infection deregulates lipid metabolism in the hepatocyte, leading to the accumulation of intracellular TG at the origin of hepatic steatosis. In addition to mechanisms involving the viral core protein in the retention and accumulation of TG [12,41], it was shown that HCV infection requires active lipid synthesis to replicate and therefore induces new metabolic needs for the cells that could be fulfilled by virus-induced lipogenesis [42,43]. Proteome analyses have shown that several CCM enzymes were upregulated in infected Huh7.5 cells but no variation in the expression of HKs was reported [44]. HKs are the first rate-limiting enzymes of glucose catabolism and, therefore, control the entry of glucose-derived carbons into the CCM. Intermediate metabolites in the CCM such as glycerol-3P and citrate for lipogenesis and pentose-P for NADPH production largely depend on glycolysis activity [45]. We show here that HCV-NS5A, through its domain 2, decreases GCK allosteric regulation and increases its activity (Figure 5 and Appendix A). This direct action of NS5A on the activity of GCK is consistent with the observed modifications of the metabolic pathways downstream of GCK such as TG accumulation and VLDL secretion (Figure 2). Thus, by interacting with GCK, NS5A modifies the intrinsic catabolic parameters of the first glycolytic enzyme to support downstream lipogenesis, which is essential for HCV replication and spreading. 

GCK is expressed almost exclusively by pancreatic β-cells and hepatocytes. This isoenzyme, unlike the other hexokinases, has a low activity at low glucose concentrations and a high activity at high glucose concentrations [46]. This characteristic allows the hepatocyte to control glucose level in the blood stream by increasing glucose uptake and glycogen synthesis when glycemia rises. A switch from GCK to HK2 isoenzyme occurs during the transition from primary to tumor hepatocytes so that HCC cell lines, such as Huh7, express HK2 but no longer GCK [22]. HK2 is active at very low glucose concentrations due to its low Km, thus allowing the cancer cells to maintain an active glycolysis independently of the glycemia to meet the important energy and biosynthesis needs for their multiplication. Moreover, unlike other hexokinases, GCK is not inhibited by its product (G6P) so that the reaction rate is driven by the supply in glucose but not by the demand for end products as it is the case for HK2 in cancer cells [47]. Our data indicate that in addition to increasing GCK activity, NS5A-D2 decreases the Hill coefficient of the enzyme from 1.6 to 1.3, reflecting a reduction in allostery (Figure 5 and Appendix A). Such a reduction in GCK allostery is also observed for natural mutants of the enzyme that are responsible for the persistent hyperinsulinemic hypoglycemia of infancy (PHHI) disease [48]. With a Hill coefficient of 1.6, normal GCK exhibits an allosteric response to physiological elevations of glycemia above 5 mM. For some PHHI mutants, GCK is maintained in a closed conformation, corresponding to the highly active form of the enzyme and the Hill coefficient falls to 1.3, making GCK highly active even at low glucose concentrations [40]. One hypothesis is thus that the interaction of GCK with NS5A-D2 induces a shift from the open inactive to the closed active conformation of GCK so that, in the presence of HCV-NS5A, GCK is active even at low glucose concentrations. This is in line with the metabolic phenotype of Huh7-*GCK^+^/HK2^−^* cells which are highly glycolytic at 5 mM glucose when NS5A-D2 is present (Figure 3D). 

HCV-NS5A is known to promote insulin resistance through IRS-1 serine phosphorylation contributing to gluconeogenesis [49]. HCV-NS5A was also described as a direct interactor of glycogen synthase kinase 3β (GSK3b), the inhibitor of glycogen synthase [26,50]. Interestingly, chemical inhibition of GSK3b prevents HCV particles assembly through perturbation of lipid metabolism [51]. Although it was suggested that the activation of glycogen synthase would be deleterious for viral replication, the consequences of HCV and HCV-NS5A on the glycogen content of Huh7 cells could not be investigated because the glycogen storage pathway is impaired in hepatoma cells. We previously reported that Huh7-*GCK^+^/HK2^−^* recovered this functionality of normal hepatocytes [22]. This study indicates that in Huh7-*GCK^+^/HK2^−^* cells expressing NS5A-D2, the intracellular content of glycogen was drastically reduced (Figure 2E), suggesting a global mobilization of glucose-derived carbon induced by NS5A-D2 in support of HCV replication. 

Our results show that cells expressing NS5A-D2 display an altered mitochondrial activity with a reduced respiratory reserve (Figure 3). The spare respiratory capacity (SRC) correlates with the metabolic adaptability of the cells in response to stress conditions [52]. The loss of SRC in cells expressing NS5A-D2 is indicative of a lower cellular metabolic flexibility. Because SRC mainly depends on the oxidation of glycolysis-derived pyruvate or fatty acids, inhibition of SRC by NS5A-D2 in Huh7-*GCK^+^/HK2^−^* cells may reflect a rerouting of carbon fluxes contributing to intracellular TG accumulation. The massive mobilization of carbon supply from extracellular glucose and glycogen to support an intense lipogenesis when the catalytic properties of GCK are modified by NS5A-D2 could be responsible for this loss of metabolic flexibility.

NS5A-D2 increases VLDL secretion and intracellular accumulation of TG in Huh7-*GCK^+^/HK2^−^* cells, highlighting the enhanced synthesis of lipids (Figure 2A–D). In addition to the induction of lipogenesis, we can speculate on the potential action of NS5A-D2 on fatty acids oxidation (FAO). Both glucose and fatty acids serve as primary catabolic substrates that provide acetyl-CoA to the TCA. The pathways of glucose and FAO are reciprocally regulated by several key metabolic intermediates and signals. Indeed, high glucose levels restrict lipid oxidation by increasing the production of malonyl-CoA, which inhibits CPT1 and, therefore, the transport of fatty acids into the mitochondria where FAO occurs (for review, see ref. [53,54]). The glucose-induced rise in pyruvate inhibits the pyruvate dehydrogenase kinase (PDK) and thereby favors dephosphorylation and activation of the pyruvate dehydrogenase complex (PDH). This results in refilling of the TCA and net export of citrate into the cytoplasm where it is cleaved by ATP-citrate lyase to oxaloacetate and acetyl-CoA. Acetyl-CoA is then carboxylated and converted to malonyl-CoA by acetyl-CoA carboxylase (ACC), the first enzyme of the fatty acid synthesis pathway. Therefore, an increased production of pyruvate resulting from the activation of GCK by HCV-NS5A could negatively regulate the fatty acid degradation process and further induce the fatty acid synthesis through citrate export from mitochondria. Overall, the increased glycolysis induced by HCV-NS5A would participate in a complex reorientation of the CCM resulting in an increased fatty acid synthesis from glucose catabolism. Further analyses of transcriptome, phosphoproteome and metabolic fluxes in cells expressing HCV-NS5A will be performed to decipher the overall metabolic consequences of glycolysis enhancement by this viral protein. 

In addition to hexokinases, HCV-NS5A interacts with cellular proteins of different pathways including the innate immune response [26,50]. In particular, the D2 domain of HCV-NS5A was previously reported to favor the infection by controlling the innate immune response [55]. This further highlights the selection by viruses of original mechanisms to control metabolism and immunity. How NS5A-induced modifications of the CCM affect the anti-HCV response in hepatocytes remains an open question. Analysis of the innate immune response upon GCK activation by NS5A in metabolically active hepatocyte such as Huh7-*GCK^+^/HK2^−^* cells should provide a better understanding of how HCV, glycolysis and innate immunity interfere. 

## 4. Materials and Methods

### 4.1. Material

Unless otherwise stated, all chemicals were from Millipore Sigma-Aldrich (Saint-Quentin Fallavier, France) and cell culture reagents were from Life Technologies (Saint-Aubin, France). Purified NS5A full-length protein, NS5A domains AH-D1 (amino acids 1 to 213 containing the amino-terminal amphipathic a-helix [AH] and domain 1), and core protein (amino acids 1 to 117) were produced with a wheat germ cell-free expression system (Cell-Free Science, Yokohama, Japan) and kindly provided by F. Penin’s team (Institut de Biologie et Chimie des Protéines, Lyon, France) [56,57,58]. Domain 2 and Domain 3 of HCV NS5A were produced in *E. coli* and purified as previously described in [39]. The sequence coding for human GCK (Variant 1_46—1395) was introduced in the bacterial expression vector pETNKI-His-3 [59]. Expression of the recombinant human GCK (residues 16—465) fused to a N-terminal hexahistidine tag was performed in *Escherichia coli* BL21(DE3) cells and using a Luria–Bertani medium. When the culture reached OD_600_~0.6 the temperature was lowered to 22 °C. Protein production was then induced with 0.4 mM isopropyl-1-thio-β-D-galactopyranoside for 23 h. Cells were harvested by centrifugation and resuspended in 40 mL of lysis buffer (50 mM sodium phosphate buffer pH 7.8, 300 mM NaCl, protease inhibitors (Complete EDTA-free, Roche Diagnostics France, Meylan, France) 10% *v*/*v* glycerol). Cells were lysed with a high-pressure homogenizer (Emulsiflex-B30) at 40 psi then centrifuged at 35,000× *g* for 45 min at 4 °C. GCK was purified by Ni^2+^ affinity chromatography (1 mL HisTrap column GE Healthcare Bio-Sciences, Uppsala, Sweden). The supernatant was loaded onto the column previously equilibrated with buffer A (50 mM sodium phosphate buffer pH 7.8, 300 mM NaCl). The column was washed with buffer A containing 20 mM imidazole and the protein was eluted with buffer A and a gradient of imidazole (20–250 mM over 8 column volumes). Fractions containing GCK were pooled and dialyzed twice against 3 L of 50 mM potassium phosphate buffer pH 7.4, 150 mM NaCl 5 mM DTT 0.2 mM EDTA. 5 mM THP and 10% *v*/*v* glycerol was added after dialysis. GCK was then concentrated with a VivaSpin 4 centrifugal concentrator (cutoff 30 kDa, Sartorius Stedim Biotech, Goettingen Germany) to 400 µM. Aliquots were flash-frozen in liquid nitrogen and stored at −80 °C.

### 4.2. Cell Culture

huh7-*GCK^+^/HK2^−^* were previously described [19] and HEK293T cells were from ATCC (CRL-3216). Cells were grown in Dulbecco‘s modified Minimal Essential Medium (DMEM) supplemented with 1 mM pyruvate, 2 mM L-glutamine, 100 U/mL of penicillin, 100 µg/mL of streptomycin and 10% Fetal Calf Serum (FCS) in 95% humidified incubator containing 5% CO2 in air at 37 °C. Culture medium and additives were from Gibco except FCS (Dominique Dutcher, Bernolsheim, France). 

### 4.3. ORF Cloning into Gateway-Compatible Plasmids 

NS5A full length or NS5A-D2 ORFs were cloned from HCV genotype 1b, isolate con1 (AJ238799), using the Gateway recombination-based cloning system [26]. Viral ORFs were PCR-amplified (with Novagen KOD polymerase, Merck-Millipore, Molsheim, France) from a DNA template using sequence-specific primers fused to attB1.1 and attB2.1 recombination sites. PCR products were subsequently cloned into pDONR223 to generate entry plasmids (BP Clonase II Enzyme mix; ThermoFisher Scientific, Courtaboeuf, France). Entry plasmids containing the ORFs of GCK and TPI1 were picked from the Human ORFeome v3.1 collection (Open Biosystem, Huntsville, AL, USA; [60]). Each ORF was transferred by in vitro recombination into the different Gateway-compatible destination vectors used in this study (LR Clonase II Enzyme mix; Life Technologies). LR reaction products were subsequently transformed into DH5α competent bacterial cells and grown overnight on LB-agarose plates containing ampicillin before amplification into liquid LB medium also containing ampicillin. Plasmids were purified using a NucleoSpin Plasmid kit from Macherey-Nagel and validated by sequencing (Eurofins, Nantes, France).

### 4.4. Cell Lines

NS5A-D2 was recombined from the corresponding entry plasmid pDONR223-NS5A-D2 into pLenti PGK Neo DEST (w531-1), a gift from Eric Campeau and Paul Kaufman (Addgene plasmid #19067; [61]). The pLenti-PGK-Neo-DEST-NS5A-D2 construct was used to produce lentiviral particles pseudotyped with VSV-G which were subsequently used to transduce 15 × 10^4^ Huh7-*GCK^+^/HK2*^−^ cells. Stably transduced cells expressing NS5A-D2 were selected in a medium supplemented with neomycin (1 μg/mL) to obtain Huh7-*GCK^+^/HK2*^—^NS5A-D2 cells.

### 4.5. Glucose and Lactate Quantification

Metabolites were quantified from cell supernatants using the Glucose Oxidase (GO) Assay Kit and the Lactate Assay Kit from Millipore Sigma-Aldrich (Saint-Quentin Fallavier, France). Assays were performed according to the manufacturer’s instructions and results were normalized to protein concentration per sample (DC Protein Assay; Bio-Rad, Marnes-la-Coquette, France). 

### 4.6. Reverse Transcription-PCR (RT-PCR) 

Total cellular mRNA was extracted using a NucleoSpin RNA kit (Macherey-Nagel, Hoerdt, France). RT was carried out on 1 µg of RNA using a High-Capacity RNA-to-cDNA kit (Life Technologies) according to the manufacturer’s instructions. PCR was run for 30 cycles with 5 µL of RT product using the GoTaq DNA polymerase (Promega, Charbonnières-les-Bains, France) in the presence of 50 pmol of primers specific for NS5A-D2 (forward primer: 5′-TTGAAGGCAACATGCACTACC-3′; reverse primer: 5′-ACGGATACTTCCCTCTCATCCT-3′) or ribosomal protein S9 (forward primer: 5′- CCGCGTGAAGAGGAAGAATG-3′; reverse primer: 5′-TTGGCAGGAAAACGAGACAAT-3′) in accordance with the manufacturer’s instructions.

### 4.7. ApoB and Lipid Quantification

ApoB concentration in medium and gradient fractions was determined by ELISA as previously described [62]. For ApoB quantifications, culture medium containing FCS was removed 24 h after seeding the cells and replaced by serum-free medium in order to measure the de novo synthesis of lipoproteins. Total TG concentration was determined with high sensitivity triglyceride fluorimetric assay from Millipore Sigma-Aldrich (Saint-Quentin Fallavier, France).

### 4.8. Iodixanol Density Gradients 

Iodixanol gradients were prepared as previously described [19]. An amount of 1 mL of culture supernatant was applied to the top of 6 to 56% iodixanol gradient and centrifuged for 10 h at 41,000 rpm and 4 °C in a SW41 rotor. Gradients were harvested by tube puncture from the bottom and collected into 14 fractions. The density of each fraction was determined with a digital refractometer HI96801 (Hanna Instruments, Lingolsheim, France). 

### 4.9. Real Time Metabolic Assays 

24 h before the assay in the Seahorse Extracellular Flux Analyzer (Agilent, Les Ulis, France), cells were seeded in XFe24- or XFe96-microplates (Agilent) pre-coated with poly-L-Lysine 0.1% (Millipore Sigma-Aldrich, Saint-Quentin Fallavier, France) at 5 × 10^4^ cells/well or 1.6 × 10^4^ cells/well, respectively. Cells were dispensed in 100 µL of DMEM medium supplemented with 10% FCS, 1 mM pyruvate, 2 mM glutamine, 1% penicillin/streptomycin (P/S). After 5 h of incubation to allow for cellular attachment, the cell culture volume was adjusted to 250 µL (XFe24) or 200 µL (XFe96) and cells were grown for 20 h at 37 °C and 5% CO_2_. Before the assay, cells were washed twice and incubated for 1 h in pre-warmed XF DMEM pH = 7.4 at 37 °C in an incubator without CO_2_ admission. The number of cells was determined at the end of the runs by automatic counting of labelled nuclei (NucBlue live cells staining; Life Technologies). The Mito Stress Tests were performed on the XFe24 Bioanalyzer, in XF DMEM pH 7.4 assay medium with 10 mM glucose, 2 mM glutamine and 1 mM pyruvate, following manufacturer’s protocol. The oxygen consumption rate (OCR) was monitored at basal level, after injection of 1.5 µM Oligomycin, 0.5 µM FCCP and 0.5 µM Rotenone/Antimycin A. The Glycolytic Rate Tests were performed on the XFe24 or XFe96 Bioanalyzers, in XF DMEM pH7.4 assay medium with 5 or 25 mM glucose, 2 mM glutamine and 1 mM pyruvate, following the manufacturer’s protocol. The extracellular acidification rate (ECAR) and OCR were measured before and after inhibition of mitochondrial respiration by 0.5 µM Rotenone/Antimycin A, and after addition of 50 mM 2-deoxy-glucose (2-DG). By measuring the amount of oxygen consumed by the cell, the contribution of mitochondria/CO_2_ to extracellular acidification is calculated and subtracted to the total Proton Efflux Rate. The resulting value is the Glycolytic Proton Efflux Rate (glycoPER). Results were normalized to cell number.

### 4.10. Coimmunoprecipitation 

NS5A was cloned by in vitro recombination (system Gateway) into pCI-neo-3xFLAG expression plasmid to be tagged with 3XFlag (thereafter referred to as Tag1) whereas GCK and TPI1 were cloned into pGluc2 to be tagged with Gluc2 (thereafter referred to as Tag2) [63]. HEK293T cells were seeded in 10 cm dishes 24 h before transfection with pCI-neo-3xFLAG-NS5A and pGluc2-GCK or pGluc2-TPI1 plasmids using jetPEI (Polyplus-Transfection, Illkirch, France). Anti-Tag1 immunoprecipitations were performed at 48 h post-transfection. Briefly, cells were harvested, washed once with PBS, and lysed in 700 µL of lysis buffer (20 mM Tris/HCl, pH 7.4, 180 mM NaCl, 1 mM EDTA, 0.5% NP40, 1% anti-protease cocktail). Cells lysates were incubated for 20 min on ice and centrifuged at 17,000× *g* at 4 °C for 20 min. Supernatants were collected, mixed with 30 µL of Anti-FLAG^®^ M2 Magnetic Beads (ref.M8823, Millipore Sigma-Aldrich, Saint-Quentin Fallavier, France), and incubated for 2 h at 4 °C. In order to limit non-specific capture of proteins, beads were saturated before use by a 2 h incubation with a homogenate of untransfected cells. Bound proteins were eluted with Laemmli buffer, denatured for 5 min at 95 °C, and subjected to immunoblotting detection using the anti-Tag2 antibody.

### 4.11. Western-Blot Analysis 

Cell lysates from 10 cm dishes were prepared in lysis buffer (1% Triton X-100, 5 mM EDTA in PBS with 1% protease inhibitor cocktail (P8340; Millipore Sigma-Aldrich, Saint-Quentin Fallavier, France). After elimination of insoluble material, proteins were quantified, separated by SDS-PAGE and analyzed by western-blot on PVDF membrane. After saturation of the PVDF membrane in PBS-0.1% Tween 20 supplemented with 5% (*w*/*v*) non-fat milk powder, blots were incubated 1 h at room temperature with the primary antibody in PBS-0.1% Tween 20 (1:2000 dilution). Incubation with the secondary HRP-labeled antibody (1:10,000 dilution) was performed after washing for 1 h at room temperature and detected by enhanced chemiluminescence reagents according to the manufacturer’s instructions (SuperSignal Chemiluminescent Substrate, Thermo Fisher Scientific, Courtaboeuf, France). 

### 4.12. Protein-Complementation Assay (PCA) 

NS5A interaction with GCK was determined by NanoLuc Two-Hybrid (N2H), a recently developed split-luciferase complementation assay [37]. In this system, two complementary fragments of NanoLuc, F1 and F2, are fused to proteins of interest. The reconstitution of NanoLuc activity by trans-complementation of F1 with F2 is dependent on the physical interaction of candidate proteins. This assay was used to test the interaction of GCK with either NS5A full-length or its D2 domain alone. The NS5A coding sequences were cloned by in vitro recombination (Gateway system; Thermo Fisher) from entry plasmids into pDEST-N2H-N1, whereas the GCK sequence was cloned into the pDEST-N2H-C2 vector. The obtained constructs were co-transfected in HEK-293T cells with the JetPrime reagent (Polyplus-Transfection). This allowed for the co-expression of NS5A N-terminally tagged with fragment F1 of NanoLuc together with GCK C-terminally tagged with fragment F2 of NanoLuc. After 48 h of culture, cells were lysed and NanoLuc activity was determined as previously described [64,65]. The bioluminescent signal obtained when co-expressing N1-NS5A and GCK-C2 was compared to the sum of the signals obtained when co-expressing N1-NS5A with F2 or GCK-C2 with F1 (background signal). 

### 4.13. GCK Activity Assay

GCK activity was determined using a method adapted from Kuang et al. [34] where glucose 6-phosphate production is measured through NADP+ reduction in the glucose 6-phosphate dehydrogenase-coupled reaction. Recombinant GCK and viral proteins used in the assay were produced and purified as described above. GCK activity was assayed in medium containing 50 mM triethanolamine (pH = 7.6), 10 mM MgCl_2_, 1.4 mM NADP^+^, with variable concentration of glucose and 1 U glucose 6-phosphate dehydrogenase (*S. cerevisiae*), equilibrated at 37 °C. The reaction was started by addition of ATP (final concentration 1.9 mM), and absorbance was continuously recorded for 30 min at 340 nm to measure NADPH production (TECAN Infinite M200). For determination of catalytic parameters of the enzyme, GCK activity was determined using a method adapted by Janke et al. [66] from Scheer et al. [67]. Briefly, 4 µL of a 1 µM GCK solution was added to 32 µL of assay buffer (100 mM Tricine/KOH, 5 mM MgCl_2_, 0.5 mM NADP^+^, 1 mM ATP, 1 U/mL glucose 6-phosphate dehydrogenase) in the presence of 0.27 to 17.2 mM glucose, on flat bottom microplates. Plates were incubated with gentle shaking for 10 min at 37 °C and the reaction stopped by addition of 40 µL of 0.5 M NaOH. Plates were sealed and incubated for 20 min at 70 °C to inactivate the enzymes. After cooling, 40 µL of 0.5 M HCl was added to neutralize NaOH in the reaction mix. The amounts of 100 µL of cycling reagent (100 mM Tricine/KOH pH 9, 8 mM EDTA, 5 mM glucose-6-phsophate, 1 mM MTT, 0.2 mM PES and 2.5 U/mL glucose 6-phosphate dehydrogenase) were added and NADPH measured by reading of absorbance at 570 nm during 30 min. The rate of reaction was calculated as the increase in absorbance per min. 

### 4.14. Statistics and Reproducibility

All the statistical analyses were performed with GraphPad Prism software. The confidence interval was set to 95% in all statistical tests. Details of statistical analyses can be found in the figure legends. The exact *p* values and sample size (*n*) are indicated either directly in the figure or in the legend. The mean ± standard error of the mean (SEM) is displayed unless otherwise stated. 

## 5. Conclusions

HCV-NS5A activates the liver specific hexokinase GCK through its D2 domain. The allosteric regulation of GCK is lost when it is bound to NS5A leading to a dysregulated glycolysis and rerouting of carbon supply and usage towards lipogenesis and VLDL secretion. As HCV replication and spreading is largely dependent on lipid metabolism, this is expected to favor virus replication and assembly, notably in the form of the highly infectious Lipo-Viro-Particles. This study describes a novel molecular mechanism supporting the metabolic reprogramming of HCV-infected hepatocytes and provide new insight into the molecular mechanisms used by viruses to control cellular glycolysis.

## Figures and Tables

**Figure 1 ijms-23-00919-f001:**
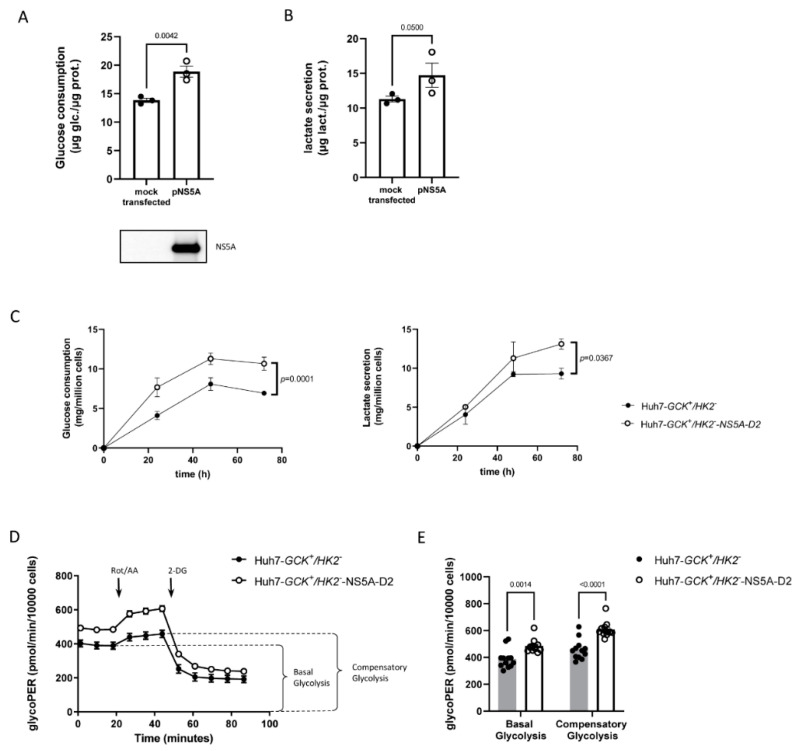
Domain 2 of NS5A increases glycolytic activity in Huh7-*GCK^+^/HK2^−^* cells. (**A**) Glucose consumption was determined in 48 h culture of Huh7-*GCK^+^/HK2^−^* cells transfected or not for HCV-NS5A (upper panel). Western-blot showing the expression of HCV-NS5A in these cells (lower panel). 30 µg of whole protein lysate was loaded on the gel for each condition. (**B**) Lactate secretion was determined in 48 h cultures of Huh7-*GCK^+^/HK2^−^* cells transfected or not for HCV-NS5A expression. Data presented means ± SEM (*n* = 3) and *p* values were determined by Student’s *t*-test. (**C**) Kinetic of glucose consumption and lactate secretion of Huh7-*GCK^+^/HK2^−^* cells expressing or not NS5A-D2 over 72 h culture. Data presented means ± SEM (*n* = 3) and *p* value was obtained from mixed-effect model analysis. (**D**) Quantification of glycolytic activity determined by the GlycoPER (proton efflux rate specific to glycolysis) using the Glycolytic Rate Test performed on a Seahorse XFe96 analyzer (Agilent). (**E**) Quantification of basal and compensatory glycolysis, determined by the Glycolytic Rate test on Huh7-*GCK^+^/HK2^−^* cells expressing or not NS5A-D2. Data presented means ± SEM (*n* = 12) and *p* values were obtained from 2-way ANOVA analyses comparing matched cell means with Sidak’s correction for multiple comparison, with α = 0.05.

**Figure 2 ijms-23-00919-f002:**
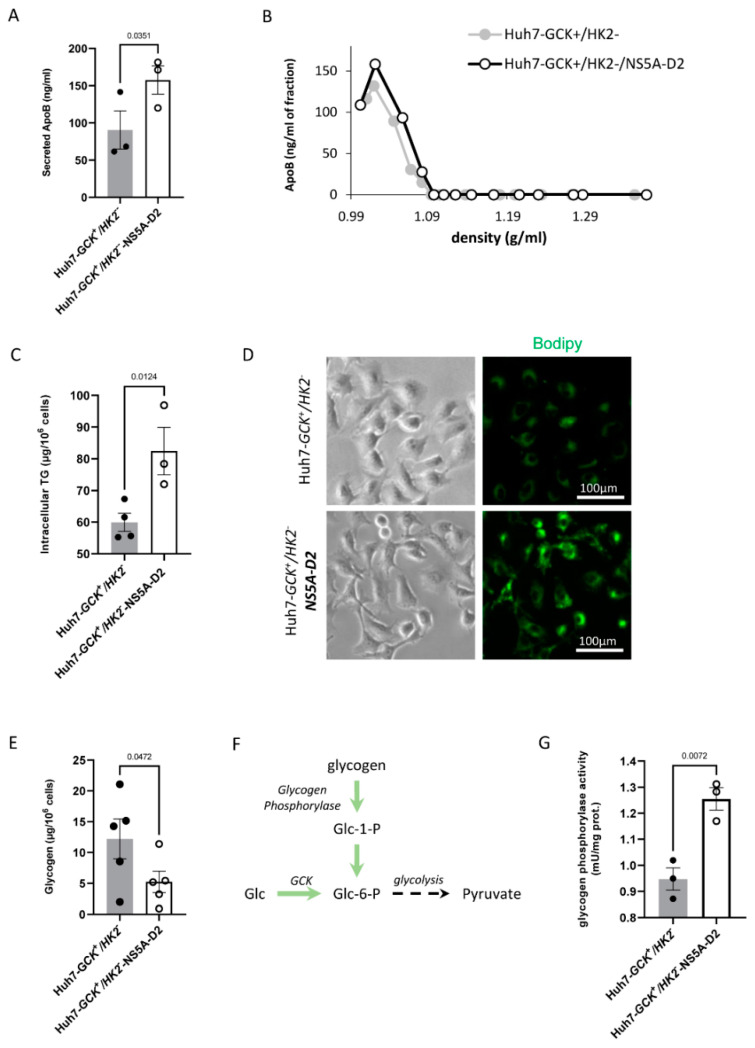
Effect of NS5A-D2 expression on ApoB secretion, intracellular TG and glycogen metabolism in Huh7-*GCK^+^/HK2^−^* cells (**A**) Apolipoprotein B secretion in supernatant after 48 h culture of Huh7- *GCK^+^/HK2^−^* cells expressing or not NS5A-D2. Data presented means ± SEM (*n* = 3). (**B**) Supernatants were analyzed by ultracentrifugation on iodixanol density gradients. Each fraction was collected and density determined before ApoB quantification by ELISA (one representative experiment). (**C**) Intracellular quantification of TG. Data presented means ± SEM (*n* = 4). (**D**) Microscopic observation of intracellular lipid droplets after Bodipy staining. (**E**) Intracellular glycogen quantification. Data presented means ± SEM (*n* = 5). (**F**) Glycogen phosphorylase activity connection to glycolysis. (**G**) Glycogen phosphorylase activity in Huh7-*GCK^+^/HK2^−^* cells expressing or not NS5A-D2. Data presented means ± SEM (*n* = 3). *p* values were determined by Student’s *t*-test.

**Figure 3 ijms-23-00919-f003:**
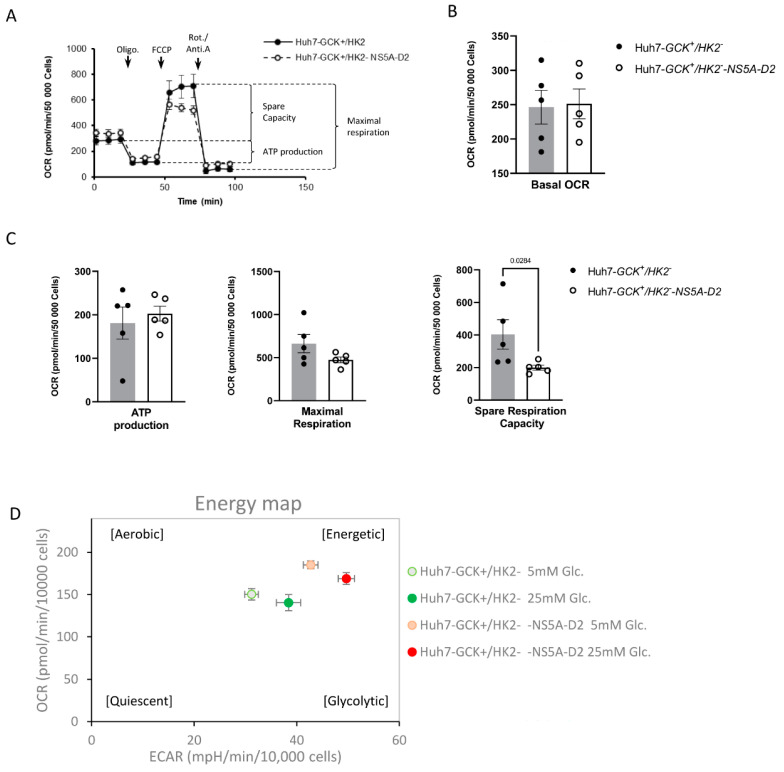
Effect of NS5A-D2 expression on mitochondrial activity of Huh7-*GCK^+^/HK2^−^* cells (**A**) Oxygen consumption rate (OCR) in Huh7-*GCK^+^/HK2^−^* cells expressing or not NS5A-D2 was determined using the Mito Stress test on a Seahorse XFe24 analyzer before and after addition of oligomycin (Complex V inhibitor), FCCP (uncoupling agent), rotenone (Rot; Complex I inhibitor) and antimycin A (Anti-A; Complex III inhibitor). (**B**) Basal OCR, (**C**) ATP production, Maximal Respiration and Spare Respiration Capacity were calculated from (**A**). Data presented means ± SEM (*n* = 5) and *p* values were determined by Student’s *t*-test. (**D**) Energy map of Huh7-*GCK^+^/HK2^−^* cells expressing or not NS5A-D2 determined from OCR and ECAR measured at basal stage in a Glycolytic Rate Test (Appendix A), performed in XF assay medium with 5 mM or 25 mM glucose. Data presented means ± SEM (*n* = 12).

**Figure 4 ijms-23-00919-f004:**
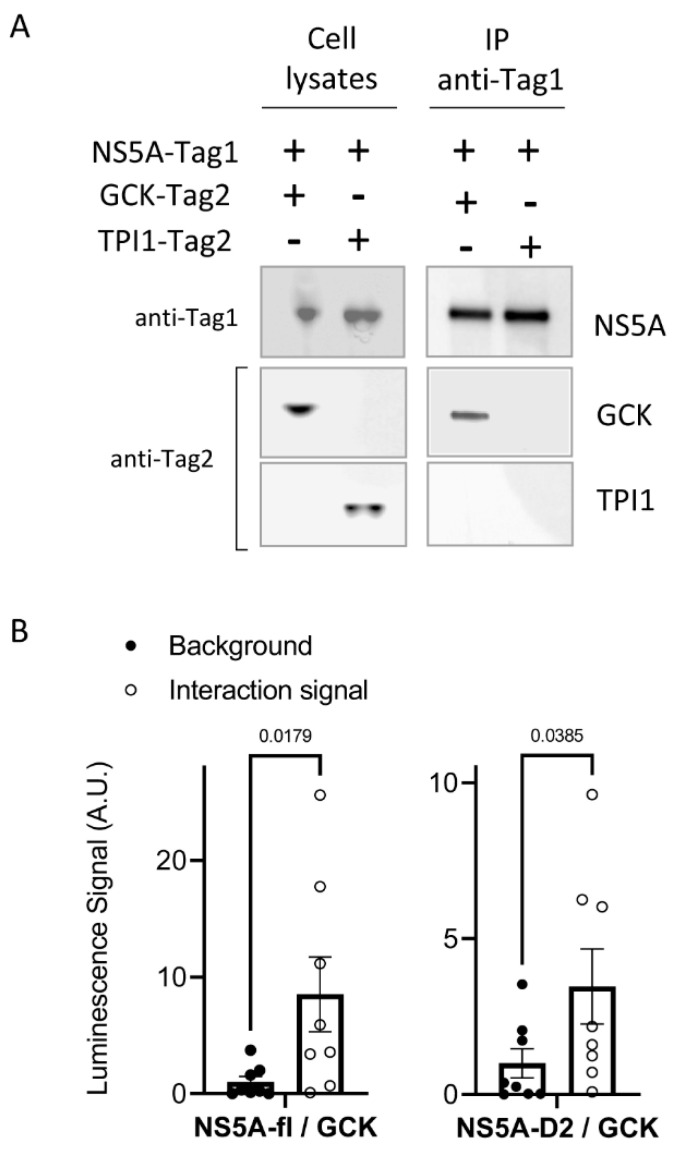
HCV-NS5A and its domain 2 interacts with GCK (**A**) Coimmunoprecipitations were performed from cell homogenates obtained 48 h post-transfection with the indicated expression plasmid for GCK, HCV-NS5A, or TPI1. Immunoprecipitations (IP) were realized using anti-3xFlag (anti-Tag1) before analysis of captured complexes by Western blotting using anti-Gluc (anti-Tag2). Expression controls of proteins in total cell lysates are presented on the left panel. (**B**) Data showing the luminescence Signal resulting from interaction of HCV-NS5A (NS5A-full length) or NS5A-D2 with GCK (open circles). See material and methods for experimental details. Background luminescence of non-specific interaction was determined in each experiment and presented (closed circles). Data presented means ± SEM (*n* = 8) and *p* values were determined by Student’s *t*-test.

**Figure 5 ijms-23-00919-f005:**
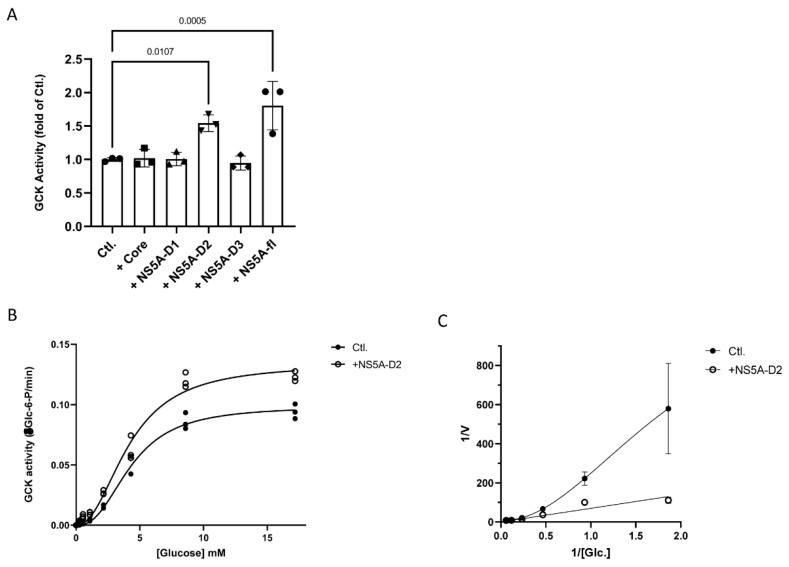
NS5A-D2 modifies catalytic parameters of GCK (**A**) Hexokinase activity was measured in the absence (Ctl.) or presence of core protein (amino acids 1 to 117), NS5A-full length, or its domains D1 (amino acids 1 to 213), D2 (amino acids 250 to 342), and D3 (amino acids 356 to 447). Activities are expressed as fold of control condition. Data presented means ± SEM (*n* = 3) and *p* values were obtained from one-way ANOVA analysis using Dunnett’s multiple comparison test with α = 0.05. (**B**) Initial velocity of hexokinase activity was measured in vitro using purified recombinant human GCK and different concentrations of glucose in the presence (+NS5A-D2) or absence (Ctl.) of purified NS5A-D2 protein. (**C**) Double reversible plot of velocity versus glucose concentration (Lineweaver-Burk representation) in the presence or absence (Ctl.) of NS5A-D2 protein. Data presented means ± SEM (*n* = 3).

## Data Availability

The data generated or analyzed during this study are included in the article and Appendix A.

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
