# Peer review of "Domain 2 of Hepatitis C Virus Protein NS5A Activates Glucokinase and Induces Lipogenesis in Hepatocytes"

_ijms, 2022, doi:10.3390/ijms23020919_

Round 1

Reviewer 1 Report

Major point

  1. The authors only showed the bodipy and lipoprotein level to for the lipogenesis. It is better to know if virus protein can also affect the genes involved in the activation of lipogenesis or inhibition of lipolysis? Or the uptake of glucose is only reason for increase of lipogenesis.
  2. Using real virus infection model to prove this concept.
  3. Instead of using liver cancer cells by replacing HK2 to HK4, using primary liver cells will have clinical relevance.
  4. Check if using GCK inhibitor can reverse the phenotype of NS5AD2

Minor points

  1. Line 44, there is need to give a brief information on acute HCV before the authors can jump into chronic stage of HCV
  2. Line 46, the Reference to be cited
  3. Line 55, for better understanding, ‘Increase influx of lipogenesis from chronic infected patient cells’ would be better
  4. Line 61, need more clarification
  5. Line 64, give an examples of other liver cell lines
  6. Line 69, should better provide the introduction of HK1-4 before talking about this cell line. Move Line 71-Line 79 up.
  7. Line 93, should be described as “the molecular basis of virus host-protein”
  8. The authors should limit the use of ‘WE’ and ‘ADDITION’ in the manuscript

Author Response

Comments and Suggestions for Authors

Major point

1. The authors only showed the bodipy and lipoprotein level to for the lipogenesis. It is better to know if virus protein can also affect the genes involved in the activation of lipogenesis or inhibition of lipolysis? Or the uptake of glucose is only reason for increase of lipogenesis.

Reply

We have very recently analyzed the metabolism of Huh7-GCK+/HK2- cells and described the metabolic changes that allow lipogenesis in these cells [1]. In this previous publication, we revealed that the expression of GCK instead of HK2 in hepatocellular carcinoma cells Huh7 modified the expression of 2932 gene transcripts. The induced modifications reorient the central carbon metabolism (CCM), stimulate mitochondrial respiration and restore essential metabolic functions of a normal hepatocyte such as VLDL secretion, glycogen storage or lipogenesis. Arguing for glucose usage as a source of carbon for lipid synthesis, we observed that in these GCK-expressing cells lipogenesis was maintained even in the absence of exogenous lipid sources, e.g. VLDL secretion was observed when the cells were cultured in the absence of fetal calf serum (FCS) but in the presence of glucose. We specifically analyzed glucose catabolism by a metabolomic and fluxomic approach in Huh7-GCK+/HK2- cells. Glucose consumption and stable isotope incorporation from [U-13C]-glucose into pyruvate were both increased in Huh7-GCK+/HK2. At the same time lactate secretion was reduced. This increased glycolytic flux together with a reduced lactate secretion account for the increased pyruvate production that essentially fuels mitochondrial TCA cycle in Huh7-GCK+/HK2 cells. Pyruvate entering the mitochondria downstream of glycolysis can be either oxidized by pyruvate dehydrogenase (PDH), producing acetyl-CoA, or converted into oxaloacetate (OAA) by pyruvate carboxylase (PC). Acetyl-CoA and OAA are then combined in the TCA cycle to form citrate. De novo lipogenesis uses citrate egress from the TCA cycle to generate cytosolic acetyl-CoA for further synthesis of fatty acids. In Huh7-GCK+/HK2 cells, we observed both an increased activity of PC and an inhibition of pyruvate dehydrogenase (PDH), consistent with a rebalanced usage of pyruvate that maintains a functional TCA cycle and supports lipogenesis. Thus, in these cells we have demonstrated that it is the reorientation of carbons from glucose, and more specifically the utilization of pyruvate produced by glycolysis, that enables lipogenesis. Since the viral protein NS5A-D2 activates the GCK enzyme and thus supports pyruvate synthesis, we assume that the enhanced glycolysis supports the lipid synthesis in Huh7-GCK+/HK2- cells.

We have undertaken a comparative transcriptomic and metabolomic analysis of Huh7-GCK+/HK2- cells expressing or not NS5A-D2. In particular, we hope to identify new potential targets for the control of lipogenesis that results from a deregulation of hepatic glycolytic flux. This study is ongoing but will not be fully analyzed and validated before 2-3 months.

The following sentence has been added to the discussion line 462-465: "Further analyses of transcriptome, phosphoproteome and metabolic fluxes in cells expressing HCV-NS5A will be performed to decipher the overall metabolic consequences of glycolysis enhancement by this viral protein."

2. Using real virus infection model to prove this concept.

Reply

Chronic HCV infection is strongly lipogenic and leads to steatohepatitis. A nonalcoholic fatty liver disease (NAFLD)-like mechanism of steatogenesis (including increased availability of lipogenic substrates and de novo lipogenesis associated with decreased oxidation of fatty substrates) is shared by all HCV genotypes [2]. The induction of lipogenesis by HCV has been observed in the liver of infected patients but also in different in vitro models of infected hepatocytes [4]–[6]. Furthermore, transgenic mice expressing the whole proteome of HCV develop hepatocellular steatosis, thus supporting a key role of HCV proteins in this metabolic reprogramming [3]. Therefore, it is well established that HCV infection induces lipogenesis and this demonstration does not need to be repeated in the present paper. The first paragraph of the introduction already presents the different mechanisms contributing to the induction of lipogenesis by HCV. The originality of our work lies in the observation that the D2 domain of HCV-NS5A is sufficient to promote lipogenesis without the requirement of other viral factors.

3. Instead of using liver cancer cells by replacing HK2 to HK4, using primary liver cells will have clinical relevance.

Reply

Although in vitro culture of primary hepatocytes is often considered as the most relevant model to study liver metabolism, it presents significant limitations. Genetic manipulations of primary hepatocytes are extremely challenging, since transfection is difficult and cells cannot be selected after lentiviral transduction. Primary hepatocytes cannot be expended in cell culture and therefore it is difficult to obtain a significant number of cells to carry out experiments. In addition, important variations from donor to donor would require repeating experiments with a large set of donors. Thus, although it would be interesting to use primary hepatocytes for our experiments, this is highly challenging and not feasible in the time frame of this review.

4. Check if using GCK inhibitor can reverse the phenotype of NS5AD2

Reply

Although it would be interesting to use a specific inhibitor of GCK to control the induction of glycolysis by NS5A-D2, this approach appears difficult to implement. Cells and especially cultured cells are dependent on glycolysis for their survival. This dependence has been widely described in the literature and has been studied for many years as a mean to induce cell death of cancer cells. To our knowledge there are no specific inhibitors of GCK, but only molecules that are analogues of glucose, such as 2-deoxy-D-glucose and interfere with all hexokinase isoenzymes. 2-deoxy-D-glucose has the 2-hydroxyl group of glucose replaced by hydrogen, so that it cannot undergo further steps of glycolysis. Indeed this molecule thus inhibits the production of glucose-6-phosphate from glucose at the phosphoglucoisomerase level (step 2 of glycolysis). Such inhibitors have been described as effective inducers of apoptosis[7]. Thus, it appears difficult to finely modulate GCK to reverse the activation due to NS5A-D2.

Minor points

1. Line 44, there is need to give a brief information on acute HCV before the authors can jump into chronic stage of HCV

Reply

Done. The sentence line 46 was modified as follow to introduce brief information on acute HCV infection:

Whereas acute HCV infection is generally asymptomatic and is followed by spontaneous viral clearance in approximately 25% of individuals, in chronically-infected patients, this metabolic rewiring contributes to pathological evolution towards steatosis, cirrhosis and hepatocellular carcinoma (HCC).

2. Line 46, the Reference to be cited

Reply

Done. The following reference has been introduced at the end of the sentence line 49.

Serfaty, L. Metabolic Manifestations of Hepatitis C Virus: Diabetes Mellitus, Dyslipidemia. Clinics in Liver Disease 2017, 21, 475–486, doi:10.1016/j.cld.2017.03.004

3. Line 55, for better understanding, ‘Increase influx of lipogenesis from chronic infected patient cells’ would be better

Reply

We modified this statement line 59 as followed:

Finally, the consumption rate of simple metabolites such as glucose increases to feed lipogenic pathways and meet viral needs [20].”

4. Line 61, need more clarification

Reply

The sentence line 63 was modified as followed:

“Indeed, the in vitro assembly of bona fide infectious LVPs was only observed in primary human hepatocytes that are functional for the production of very low-density lipoproteins (VLDLs).

5. Line 64, give an examples of other liver cell lines

Reply

The sentence line 66 was modified as follow and a reference has been added. “The HCC cell line Huh7 that is widely used to study HCV replication in vitro is defective for this function alike other HCC cell lines such as HepG2 [21].”

6. Line 69, should better provide the introduction of HK1-4 before talking about this cell line. Move Line 71-Line 79 up.

Reply

Done. The paragraph is now written as followed:

Hexokinases control the first rate-limiting step of glucose catabolism by phosphorylating glucose to glucose-6-phosphate (G6P), fueling glycolysis as well as glycogen, pentose phosphate and triglyceride synthesis. The human genome contains five genes encoding distinct hexokinase isoenzymes, named HK1, HK2, HK3, HK4 (also named GCK for glucokinase) and HKDC1, with distinct enzymatic kinetics and tissue distributions. While HK2 is active at low glucose concentration, GCK has an allosteric catalytic comportment with a weak activity at low glucose concentration and a strong activity at higher concentration. Thus, GCK contributes to the regulation of glycemia by controlling hepatic glycolytic activity. During carcinogenesis of hepatocytes, GCK is replaced by HK2 so that HCC cell lines express HK2 instead of GCK. To circumvent this limitation, we recently developed a metabolically active Huh7-derived cell line (Huh7-GCK+/HK2-) by replacing the cancer-type hexokinase HK2 by GCK [23]. We restored the expression of GCK and knocked-down HK2 in Huh7 to generate the Huh7-GCK+/HK2- cell line. We observed that HK2 knockdown and GCK re-expression rewired central carbon metabolism (CCM), stimulating mitochondrial respiration and restoring essential metabolic functions of normal hepatocytes such as lipogenesis, VLDL secretion and glycogen storage [23]. Altogether, this makes of the novel Huh7-GCK+/HK2- cell line a functionally relevant model to study HCV interaction with glycolytic, glycogenic and lipogenic pathways.

7. Line 93, should be described as “the molecular basis of virus host-protein”

Reply

The sentence line 95 has been replaced by the following:

By analyzing these interactomes, key metabolic pathways for viral replication were identified [27–30], but only a few studies have specifically described interactions between viral proteins and host enzymes that directly modulate their activity [31–34].

8. The authors should limit the use of ‘WE’ and ‘ADDITION’ in the manuscript

Reply

We have taken this comment into account and amended the text accordingly.

Reviewer 2 Report

The manuscript by Perrin-Cocon et al used the newly developed Huh7-GCK+/HK2- cell line to study the consequences of HCV-NS5A expression on glucokinase (GCK) and lipogenesis in hepatocytes. This study reported that HCV-NS5A activates the GCK of hexokinases through its D2 domain. The is a well-designed and the data presented in this study are convincing.

This reviewer finds the manuscript a great addition to the existing knowledge on HCV pathogenesis.

Few minor comments are given below:

  • Please add the schematic diagram of HCV-NS5A protein with its core protein and all domains with the reported functions in the manuscript. A diagram will be helpful for the readers to understand the domains of the NS5A protein.
  • Figure 1: Figure 1 A and B can be presented as supplemental figure as these outcomes are known from the previous study.
  • Line 105: Add Figure 1 A, B or according to the new figure numbers.
  • Line 112: Change the word ‘like’ to ‘similar to’. Figure 1 a, b could be done with the Huh7-GCK+/HK2- cells transfected with NS5A-D2.
  • Replace ‘Are presented means+SEM’ with ‘data presented means+SEM’ in the figure legends where applicable.

Reviewer 3 Report

The present study evaluates NS5A-D2 can reprogram central carbon metabolism towards a more energetic 30 and glycolytic phenotype compatible with HCV needs for eplication. The presented studies will contribute to understand of how HCV, glycolysis and innate immunity interfere. The research subject is interesting and brings scientific important data in the field. However,There are some changes of the manuscript points in the paper that need to performed.

1. At line 110–114, should the results shown in Figure 1A be presented firstly ? An internal reference belt is needed in Figure 1A.

2. Figure 2E: Horizontal coordinate annotation bold should be uniform.

3. Figure 4A needs to be improved.

4. Discussions should highlight novelty and originality, compared with similar studies.

5. Conclusions should be added.

6. Please update the reference.

7. At line 398, … 46-1395 … should be … 46–1395 ….

8. At line 399, … 16-465 … should be…16–465 ….
